# A Review on Metastable Silicon Allotropes

**DOI:** 10.3390/ma14143964

**Published:** 2021-07-15

**Authors:** Linlin Fan, Deren Yang, Dongsheng Li

**Affiliations:** 1State Key Laboratory of Silicon Materials and School of Materials Science and Engineering, Zhejiang University, Hangzhou 310027, China; msefll@zju.edu.cn (L.F.); mseyang@zju.edu.cn (D.Y.); 2Cyrus Tang Center for Sensor Materials and Applications, Zhejiang University, Hangzhou 310027, China

**Keywords:** metastable phase, silicon, allotropes, synthesis

## Abstract

Diamond cubic silicon is widely used for electronic applications, integrated circuits, and photovoltaics, due to its high abundance, nontoxicity, and outstanding physicochemical properties. However, it is a semiconductor with an indirect band gap, depriving its further development. Fortunately, other polymorphs of silicon have been discovered successfully, and new functional allotropes are continuing to emerge, some of which are even stable in ambient conditions and could form the basis for the next revolution in electronics, stored energy, and optoelectronics. Such structures can lead to some excellent features, including a wide range of direct or quasi-direct band gaps allowed efficient for photoelectric conversion (examples include Si-III and Si-IV), as well as a smaller volume expansion as lithium-battery anode material (such as Si_24_, Si_46_, and Si_136_). This review aims to give a detailed overview of these exciting new properties and routes for the synthesis of novel Si allotropes. Lastly, the key problems and the developmental trends are put forward at the end of this article.

## 1. Introduction

Diamond cubic silicon lies at the heart of integrated circuits and the photovoltaic and electrochemistry industry thanks to its distinct properties illustrated in Figure 1 [1,2], providing a convenient lifestyle for humans. Despite this, it is not the best material with the most desired performance, and it has encountered bottlenecks on the way forward; thus, scientists are beginning to consider whether there are some functional silicon materials to satisfy the ever-increasing demand. The phase diagram of Si offers a good idea; silicon can exist in different structures and these novel forms of Si offer exciting prospects to create Si-based materials with properties tailored precisely toward specific applications.

Low power consumption, high density, ultra-fast response, and processing rate are the forward aims of information materials all the time. Under the guidance of “Moore’s Law”, problems occur with the feature size of integrated circuits shrinking, such as RC delay, Joule heat, electromagnetic interference, and quantum tunneling. Therefore, researchers proposed a road to “beyond Moore”, photoelectric interconnection technology, using photons instead of electrons to transfer information to overcome the inability mentioned above [3,4]. Although it is easy to realize photoelectric interconnection through III–V semiconductors, if silicon has the desirable properties, it will become reliant on the earth-abundant storage and mature industrial processing methods. However, diamond silicon (DC-Si, Si-I, *cF*8) is an indirect band gap semiconductor, and its low light absorption and emitting efficiency limit its further application in optoelectronic devices. Silicon allotrope is a burgeoning branch, which may exhibit efficient optical absorption and emission far beyond Si-I; for example, Si-IV (HD-Si, *hP*4) shows an intense visible emission at about 1.5 eV and a near-infrared emission at 0.8 eV in Figure 2a [5,6].

In 1954, Bell Labs manufactured the world’s first practical silicon single-crystal solar cell, which triggered a boom in solar cell research. In terms of cost and commercialization, silicon is the material of choice for solar cells, although the efficiency of crystalline silicon cells is still too low compared to III–V solar cells and multijunction cells. Exotic silicon allotrope materials optimized for solar energy conversion have higher carrier mobility and stronger visible light absorption capacity due to the direct gap band and multiple exciton generation effect, so that they can take a place in the photovoltaic industry in the near future [7,8,9,10,11].

**Figure 2 materials-14-03964-f002:**
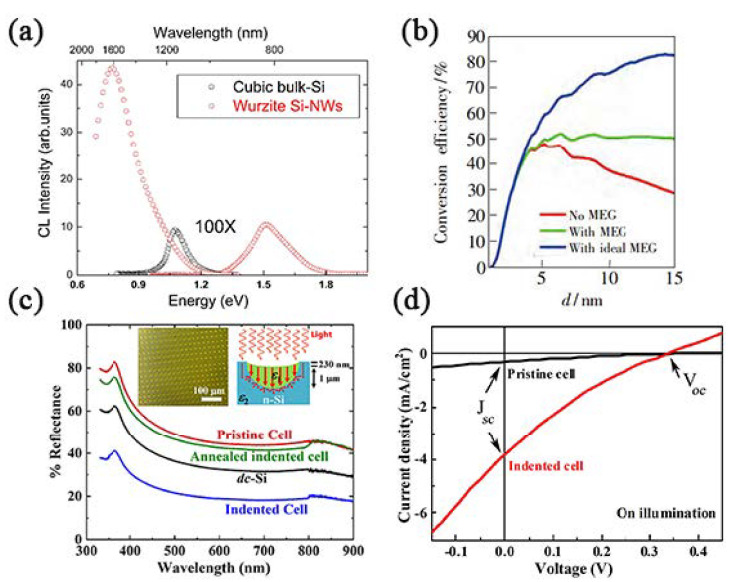
Outstanding performances of silicon allotropes in solar cells. (**a**) Comparison between CL spectra of 2H-Si nanowires and standard cubic silicon. Si nanowires have two light-emitting peaks at about 1.5 eV and 0.8 eV. Reprinted with permission from Ref. [6]. Copyright 2014, *Sci*. *Rep*. (**b**) Calculated power conversion efficiencies of single stage Si(*cI*16) quantum dots solar cells at solar spectra AM1.5 G. Adapted with permission from Ref. [10]. Copyright 2018, Chinese Journal of Luminescence. (**c**) Reflectance spectra of the standard Si wafer, pristine n^−^p-p ^++^ solar cell wafer, and the indented cell before and after annealing. The inset shows the optical image of indents made on cell and schematic shows the enhanced light absorption by the indented area. The symbols ε_1_ and ε_2_ indicate the dielectric functions of the Si(*hR*24) and Si(*cF*8), respectively. (**d**) The J–V characteristics of pristine cell (black) and indented cell (red). Adapted with permission from Ref. [11]. Copyright 2020, Scripta Materialia.

Silicon is the most prospective electrode material for lithium/sodium-ion batteries, whose theoretical specific capacity can reach 4200 mAh·g^−1^ (Li_22_Si_5_) [2]. However, its sensitive volumetric expansion as high as 300% during cycling is the main obstacle to its commercialization, which will cause a security issue and large capacity loss. From Figure 3, we can see that some open framework allotropes of silicon, such as Si_24_, Si_136_, and Si_46_ exhibit high tolerance to volume change, since the open, simple nanocage-based structures with strong covalent bonds allow for guest atom intercalation. The theoretically predicted capacities of Si_24_, Si_136_, and Si_46_ are 159, 168, and 791 mAh·g^−^^1^ respectively; however, the experimental data are higher than the theoretical capacity, and there is a maximum corresponding to the acceptable overlithiation before structure damage [12,13,14].

Silicon possesses a complex free-energy landscape with rich local minima; hence, a multitude of allotropes can exist similar to carbon. Figure 4 is a pressure–temperature phase diagram of silicon that clearly illustrates the stability range of each high-pressure phase [15,16]. From it, we can know that the diamond cubic structure is the thermodynamic ground state of silicon in room temperature and atmospheric pressure conditions, and the crystal undergoes six phase transitions, consisting of six high-pressure single phases (Si-II, Si-XI, Si-V, Si-VI, Si-VII, and Si-X) during the process of increasing the pressure from normal pressure to 105.2 GPa at room temperature. It is worth noting that the coordination number of silicon crystals continues to increase as the pressure rises, resulting in an enhancement in the degree of electron delocalization and metallicity [7]; thus, all these high-pressure phases above 12 GPa are metals. Because the Gibbs free energy of high-pressure phases are high, they can only exist stably under certain temperature and pressure, and a reversible phase change will occur on standard decompression. Specifically, Si-I can directly transform to Si-II, but not vice versa, requiring a series of processes and phase changes (Si-II→Si-XII→Si-III→Si-IV→Si-I) [7]. If the rate of decompression is high enough, the appearance of Si-VIII (*tP*32) and Si-IX phases can be observed [7,17]. In addition to these phases attained from high-pressure treatment, the experimental observation of ST12 and BT8 induced by ultrashort laser pulses is reported [18]. Of course, not all allotropes can be obtained directly from Si-I. Some low-density clathrates, such as Si_136_, need to be prepared by thermal decomposition of the Zintl phase [19]. Figure 5 is schematic overview of silicon structures, as well as a summary of structural changes under different treatments. Overall, atomic arrangement is closely related to a material’s properties. It is, thus, highly believed that the research on silicon allomorphs will bring broad prospects to silicon-based materials.

As mentioned above, silicon allotropes have many types and various properties, representing a feasible opportunity to address the ever-increasing demand. It is worth emphasizing that the stability at ambient temperature and pressure is the key to its further wide application. Therefore, the aim of this review is to elucidate these exciting new properties and avenues for the synthesis of novel metastable Si. First, the common metastable structures of silicon are reviewed in Section 2, involving their history, properties, and mutual relationships. Next is a detailed summary of the pathways to exotic metastable silicon allotropes in Section 3. Lastly, the key challenges and developmental prospects for the field are discussed.

## 2. Known Metastable Phases of Silicon

From the perspective of topology, the possible structures increase exponentially with the number of atoms; however, how many silicon allotropes could potentially remain stable in ambient conditions? To date, the known silicon allotropes stabilized in ambient conditions include Si-I and the most common metastable phases (Si-III, Si-IV, Si-XII, Si-VIII, Si-IX, BT8-Si, ST12-Si, Si_24_, Si_46_, and Si_136_ in Figure 5). Below, we discuss these typical metastable phases around their structure, performance, development history, and promising applications. The basic information of each metastable phase is given in Table 1. Before proceeding, we note that marking symbols for structures vary widely in different literatures, and the Pearson symbol (as shown in the third column of Table 1) is adopted herein.

### 2.1. The Metastable Phases under Slow Decompressure Rate (Si(cI16), Si(hR24), and Si(hP4))

As mentioned above, Si-II (*I*4_1_/*amd*, *tI*4) cannot directly convert to Si(*cF*8), whereas Si(*cI*16), Si(*hR*24), and Si(*hP*4) polymorphs are largely recoverable under slow pressure release starting from Si(*tI*4); thus, we put these three phases together for discussion.

Back in 1962, Minomura and Drickamer [27] accidently found that, when a pressure up to 200 kbar was applied at room temperature, the resistivity of silicon dropped to 10^−5^ ohm/cm, similar to metallic materials. They concluded that the change in resistivity and the rate of change were related to the shear force applied to the silicon, i.e., a larger shear force led to faster resistivity changes. One year later, Wentorf and Kasper [28] first discovered two new forms of silicon (Si(*cI*16) and Si(*hP*4)) from the same high-pressure experiment.

Si-III (also known as BC8-Si, Kasper phase, *Ia*$\bar{3}$, *cI*16) of a cubic structure shows the tetrahedral surroundings of a contiguous pair of atoms along the body diagonal, which is fourfold coordinated with one Si–Si distance at ∼2.37 Å but the other three at ∼2.38 Å [29]. The Raman peaks of Si(*cI*16) were around 160, 380, and 438 cm^−^^1^ [30]. However, there remain numerous controversies regarding the width of the band gap. An early study by Besson suggested that Si(*cI*16) is a hole semimetal with low charge carrier density [31], but Cohen predicted Si(*cI*16) as a semiconductor with a theoretical direct band gap of 0.43 eV. Later, the observed temperature variation of conductivity was believed to conform to the expected behavior of a narrow gap semiconductor with the band gap of about 30 meV [31], and the data were verified through diamond anvil cell (DAC) [29,32], nanoindentation [16], and wet chemistry experiments [33]. The narrow-gap semiconductor of Si(*cI*16) is a highly desirable feature for multiple exciton generation-based solar cells, and the simulated maximum energy conversion efficiency can reach 51.6% due to effective usage of high-energy photons (Figure 2b) [34,35]. Another confusing phenomenon is that the reported conductivity values for Si(*cI*16) are orders of magnitude greater than Si(*cF*8), but no indication of superconductivity can be found even down to 1.2 K.

As for Si-IV (also called HD-Si, hex-Si, lonsdaleite Si, *w*-Si, *P*6_3_/mmc, *hP*4), the results of XRD could be indexed for a hexagonal cell (a = b = 0.380 nm, c = 0.628 nm), corresponding to a structure related to the wurtzite phase [36]. Although Raman scattering is one of the most effective methods for detecting Si(*hP*4), the precise peak position was slightly shifted in different reports. In Shukla’s experiment, Raman active optical modes of Si(*hP*4) were around 515 cm^−1^, 507 cm^−1^, and 495 cm^−1^, respectively [23]. Rodichkina and Lysenko [24] reported a reversible photo-induced formation of Si(*hP*4) in initially metal-assisted chemical etching-fabricated diamond cubic Si nanowires, but they observed the Raman peaks centered near 490 cm^−1^, 510 cm^−1^, and 517 cm^−1^, corresponding to the Si(*hP*4) phase in their experiment. Theoretical and experimental studies suggest that the Si(*hP*4) indirect band gap is in the near-infrared range, near the emission of erbium in bulk silicon [6,23], while a direct transition at point Γ is at around 1.4–1.6 eV [37,38,39,40] (see in Table 2) and the emission efficiency is 2–3 orders of magnitude greater than that of Si(*cF*8) in the visible-light region. Surprisingly, Si(*hP*4) is transformed into a direct band gap semiconductor under strain (>4%) [40] or germanium doping [41], and its band gap width shows good tunability for different strain and compositions; hence, it may open a new scenario in the field of silicon-based rare-earth-free optoelectronic devices.

Si-XII(R8, *R*$\bar{3}$, *hR*24) was first discovered in 1994 by Crain [42]. The reversible phase transition study implied that Si(*tI*4) transforms first to Si(*hR*24) before converting to the Si(*cI*16); thus, these three structures are closely related. The most prominent Raman vibrational frequency and the band gap of Si(*hR*24) were given as 350 cm^−1^ and ~240 meV in the experiment. Si(*hR*24) has a unique form among these three metastable silicon phases in that it contains five-membered rings in a rhombohedral unit cell, the presence of which may account for the special electronic abilities of the material including high carrier mobility and electrical activation at room temperature [43,44]. Taking advantage of the high carrier mobility and a ~50% lower reflectance, Mannepalli et al. [11] made a silicon solar cell with an ordered dotted Si(*hR*24) array of a few hundred micrometers on the top layer, which showed ~10-fold improvement in the photocurrent density (Figure 2c,d).

Whether it is for devices or materials, stability is a crucial indicator. For the above three allotropes, we can find that the thermal stability range of each phase is diverse, even if the phase-stable temperature of the same phase synthesized by different methods is different. For example, at ambient pressure, Si(*hR*24) synthesized through HPHT undergoes an irreversible transition to Si(*hP*4) at 200 °C. In another article, Ganguly et al. [33] reported high-thermal-stability Si(*cI*16) nanoparticles prepared by a colloidal route, whose the characteristic peaks remained when heated to 650 °C in a tube furnace under the flow of argon for 2 h. Compared with other metastable phases, Si(*hP*4) shows relatively higher thermal stability and, upon further annealing to 500 °C or even higher temperature, Si(*hP*4) transforms back to the stable Si(*cF*8) [45,46]. In a recent study, the authors [47] systematically presented the thermal evolution of the indentation-induced phases of silicon and offered an updated schematic of the changing phase composition during annealing. From Figure 6, we can know that Si(*hR*24) can be rapidly transformed to partially stable Si(*tP*12/*tP*20) in a narrow temperature range, and then Si(*tP*12/*tP*20) and Si(*cI*16) fully anneal out at 240 °C, transforming to Si(*hP*4). The thermal stability limits the operating temperature of the device; however, from another perspective, this may be a good way to synthesize required pure phase silicon, taking advantage of the thermal evolution of these metastable phases.

### 2.2. The Metastable Phases under Fast Decompressure Rate (Si(tP32) and Si-IX)

As stated in the introduction, the Si(*tP*32) and Si-IX phases are obtained following rapid release of pressure after static compression. Specially, Si(*tP*32) involves t32(*P*$\bar{4}$2_1_*C*) and t32*(*P*4_3_2_1_2), i.e., two different structures, which contain 32 Si atoms per unit cell and can be characterized by different arrangements of spiral chains [20]. Zhu et al. [48] predicted that the t32 structure is a quasi-direct gap semiconductor of 1.28 eV using HSE06, making it promising for photovoltaic applications. In addition to a diamond anvil cell experiment [17], the evidence for the metastable Si(*tP*32) phase has been reported within laser-driven shock experiments [18,49]. Akio Hirose’s team [49] prepared Si(*tP*32) grains of about 243 nm using femtosecond laser-driven shock wave and confirmed that the XRD peak agrees well with the (423) plane of the metastable Si(*tP*32) phase.

Similarly, the Si-IX has three possible candidate structures, *Ibam* [50], *P*4_2_/m, and *P*$\bar{4}$ [51], and it could even be a mixture. So far, the nature of Si-IX has remained mysterious, and the only certainty is that Si-IX consists of large tetragonal unit cells with 12 silicon atoms. Goswami [52] prepared Si-IX grains with a size range from 2 to 5 nm using the propagation of the shock wave during thermal spray. They believed that the formation of a metastable phase depends on how quickly it is quenched to room pressure, determining how far is it from the equilibrium condition, whose role is similar to the decompression rate in the diamond anvil cell method. It is also worth noting that the Si-IX shares many similarities with Si(*cI*16) and is believed to be either a semimetal or a narrow band semiconductor within theorical prediction. Another application of many-valley phonon scattering could be of interest as a superconductor if it were doped [50]. 

### 2.3. Low-Pressure Tetragonal Allotropes (Si(tI16) and Si(tP12))

In 2015, Rapp and his coworkers [18] produced BT8-Si(*I*4_1_*/a*, *tI*16) and ST12-Si(*P*4_3_2_1_2, *tP*12) within bulk silicon for the first time, leading to further studies on shock waves induced by femtosecond lasers. 

Si(*tI*16) has 16 atoms in the body-centered tetragonal unit cell with the space symmetry group $C_{4h}^{6}$-*I*4_1_*/a*, and its density is 2.73 g/cm^3^ (17% higher than Si(*cF*8)) [53]. The calculated result indicates that the electrical structure of Si(*tI*16) is similar to Si(*cI*16); hence, it is likely that Si(*tI*16) is a narrow gap semiconductor, indicating its usage for improved mid-infrared detection and for multiple exciton generation, with applications in next generation photoelectric [18].

The ST12 structure with a unit cell of 12 atoms was first discovered by Cohen in the process of decompression from the *β*-Sn phase in germanium, and Si(*tP*12) was also experimentally observed in silicon [18]. According to Rapp’s calculations, Si(*tP*12) has a relatively low density of 2.47 g/cm^3^ (6% more dense than Si(*cF*8)), and it is expected to be thermally stable to above 400 °C [18]. Moreover, the calculated density of states illustrates that it is an indirect band gap semiconductor with a band gap energy 1.67 eV for empirical pseudopotentials [44] and 1.1 eV for DFT/LDA [37], respectively, but might perform fascinating superconducting behavior when sufficiently doped [44].

### 2.4. Low-Density Si Clathrates (Si(cF136), Si(oC24), and Si(cP46))

The important progress on low-density Si clathrates has been well discussed in the previous reviews [54,55]. Such open-cage allotropes serve as diffusion channels and reservoirs for ions, with suitable shape and size. Furthermore, the theoretically predicted band gap is wider than that of diamond cubic silicon, illustrating a greater overlap of absorption with the solar spectrum [56]. Here, we supplement this finding with some research work from the last five years.

Both Si(*cF*136) and Si(*oC*24) belong to type II clathrate structures, and they can be acquired from metastable precursor (Na_4_Si_4_) under 0.1 MPa and 8 GPa, respectively [57,58]. O’Keeffe found the type II structure to have the lowest energy among all clathrates using empirical potentials [59]. The discovery of the Na_x_Si_136_ clathrate in the 1960s provided the first demonstration that silicon can exist in expanded, low-density forms [24], and the new allotrope of silicon, Si(*cF*136), was yielded in their work for five years afterward [60]. The cage-like structure Si(*cF*136) possesses 136 Si atoms per unit cell in *sp*^3^-like arrangement, which consists of five- and six-membered rings [61]. The Raman scattering experiment showed peaks of Si(*cF*136) at 120 cm^−1^ and 490 cm^−1^ at the Γ point and the most prominent peak appeared around 487 cm^−1^ [20]. Theoretical calculations and experiments about electrical structure were performed on Si(*cF*136), with the consensus that it has a wide optical band gap of 1.9 eV, approximately two-fold wider than 1.12 eV [19,62]. The photovoltaic response was observed in the Si(*cF*136) film although short-circuit current, open-circuit voltage, and conversion efficiency were still not ideal (9.2 nA, 0.036 V, and 10^−5^%) [63]. Li et al. [13] reported that the discharge capacity (181.2 mAh·g^−1^ at room temperature and 246.9 mAh·g^−1^ at 45 °C) is higher than the theoretical capacity of Si(*cF*136) to form Na_24_Si_136_ (168 mAh·g^−1^), which is attributed to the formation of fcc crystals (Figure 3b). It is worth mentioning that Si(*cF*136) can remain stable up to nearly 600 °C (even higher thermal stability than Si(*oC*24)) [64], which leads to further applications in PV and batteries.

In 2015, Duck Young Kim [57] used a two-step method to synthesize Si(*oC*24) in the Na–Si binary system. The orthorhombic structure of Si(*oC*24), which has 24 Si atoms per unit cell, contains open channels along the crystallographic a-axis that are formed from six- and eight-membered *sp*^3^ silicon rings [57]. Such a structure is suitable for battery anodes for ion storage and diffusion; not only was this material reported to be structurally and chemically stable to above 450 °C, but it was also found to have a high reversible stable capacity. Furthermore, the observed absorption edges of Si(*oC*24) are at 1.29 eV and 1.39 eV, representing indirect and direct transitions, respectively [57], and the negligibly small difference makes Si(*oC*24) a promising material for solar conversion applications and light-emitting devices.

Type I silicon clathrate Si(*cP*46), which consists of Si with a regular arrangement of 20-atom and 24-atom cages connected together through five-atom pentagonal rings, has a simple cubic structure (a = 10.335 Å) with 46 Si atoms per unit cell [14]. Beyond our expectation, the guest-free Si(*cP*46) clathrate could not be obtained via a ‘degassing’ process, but only by a solution process which involved soft oxidation of Na_4_Si_4_ by a Hofmann-type elimination–oxidation reaction scheme because its cages were too small to remove Na atoms completely [14]. The capacity of the half-cells made from the mixture of Si(*cP*46) and byproducts underwent an increase at 50 cycles and then a decrease from 809 to 553 mAh·g^−1^ after 1000 cycles, while maintaining 99% Coulombic efficiency. The previous stage long-term stable date (809 mAh·g^−1^) occurred after the insertion of 48 Li atoms; the decrease in capacity, however, corresponded to overlithiation beyond 66 Li atoms that caused irreversible damage to clathrate [14,65].

## 3. Pathways to Exotic Metastable Silicon Allotropes

The allotropic structures of silicon have been widely studied for nearly 60 years, and more than 10 metastable phases of silicon have been experimentally observed [7]. In fact, theoretical calculations on the properties and structure of polymorphs have far exceeded experimental preparations, but they can provide powerful guidance for synthesis in the laboratory. Clearly, it is nonlinear, nonstandard conditions very far from thermodynamic equilibrium in a confined environment that enable the synthesis of novel functional structures of metastable silicon. Next, we review research progress in synthetic means of exotic forms of silicon. In general, there are two main research ideas for the preparation of novel phase silicon (Figure 7). One is a top-down method, applying an external force on silicon materials to cause a phase transformation through physical or chemical processes; the other, known as a bottom-up method, is a process in which silicon precursors are assembled into silicon materials with specific structures and properties through a chemical reaction.

### 3.1. Top-Down Approach

#### 3.1.1. Diamond Anvil Cell Method (DAC)

The DAC method is the earliest and the most conventional technology to get new silicon phases, and almost all high-pressure phase silicon can be prepared under different stress levels or decompression rates. Furthermore, researchers evolved the high-temperature/high-pressure method (HTHP) and chemical high-pressure routes on the basis of a simple diamond anvil cell (HP). The equipment of DAC is easy to achieve, which is sufficient for the study of phase change mechanisms and new crystal phases, but not suitable for applications that require large size or high integration.

Besson [31] reported for the first time that pure phases of Si(*hP*4) and Si(*hR*24) phases were prepared under hydrostatic pressure in the DAC method. When the pressure was increased to 12 GPa before returning to room pressure, the Si(*cF*8)→Si(*tI*4)→Si(*cI*16) transitions took place in the microcrystalline phase, whereas Si(*cI*16) changed into pure phase Si(*hP*4) after annealing at 470 K for 2 h. In 2016, Smith [66] carried out the same study on silicon nanowires prepared by metal-assisted chemical etching (MACE). However, Raman measurements indicated that Si(*hP*4) was the dominant phase in the recovered silicon nanowires without further annealing. Comparing the two experiments above, we speculate that the variance in morphology led to a difference in the stress distribution in the material, and the stress distribution on the nanowires was uneven.

Silicon morphology has an impact on structural evolution, as does size. Recently, Zeng’s team [67] studied the origin of plasticity in a large number of randomly oriented Si nanoparticles using in situ high-pressure radial X-ray diffraction, allowing not only to observe the changes in the crystal structure and orientation relationship of these particles with stress, but also to quantitatively obtain the elasticity and plastic deformation with stress, and realize its direct correlation with structural changes. Interestingly, they found a large nanoparticle transition to Si(*tI*4) while small nanoparticles transitioned to Si(*hP*1) after exceeding the critical pressure, which also identified the size as a significant parameter of the lower-energy structure.

Exactly speaking, the raw material of the chemical high-pressure method is Zintl rather than silicon; thus, it is a bottom-up solid phase method. Zintl phases have structural features that are similar to the desired product, lowering activation energies for transformation and facilitating topotactic reactions; therefore, they can provide a platform for the discovery of novel silicon. Kim et al. [12] applied a novel two-step synthesis methodology to prepare Si(*oC*24) and provided a potential route to new silicon clathrates. First, the Na_4_Si_24_ precursor was synthesized at high pressure; second, the sodium guest was removed from the precursor by a thermal ‘degassing’ process. At present, there have been several reports about using this method to synthesize Si(*oC*24) [12], Si(*cF*136) [58], and Si(*hP*4) [32]. Compared with the HPHP and direct HP methods, the phase transition pressure is lower than that in the chemical HP method under the same conditions, but the test results of binary system are not significantly different from the properties of the metastable phase prepared by the monolithic silicon system. Recently, Courac et al. [68] carried out a detailed study of its high-pressure/high-temperature phase diagram and thermodynamics and kinetics behaviors, which contributes to the use of Na_4_Si_4_ for high-pressure syntheses.

#### 3.1.2. Indentation Method

Indentation is a method of applying a localized pressure to a material via uniaxial point loading. Although both DAC experiments and indentation experiments apply stress directly to the materials, there are a few distinct differences. We outline three factors underlying the kinetic differences between indentation and DAC. First, Si(*hR*24) remains stable at ambient pressure after indentation, whereas it is unstable after a DAC. Recently, Wong et al. [69] produced an Si(*hR*24)-dominant Si material (the direct XRD result suggested Si(*hR*24) as high as 70%) via close-packed indentation arrays, and explained that the indent contained an intrinsic compression of ∼4 GPa, which stabilized the Si(*hR*24) phase. Second, the rapid decompression from Si(*tI*4) formed under the indentation tip leads to the formation of pure amorphous Si(*a*-Si) instead of Si(*hR*24) and Si(c*I*16) [69]. This clearly suggests that the nucleation of Si(c*I*16)/Si(*hR*24) is inhibited during indentation but the exact reason remains unknown. Third, the high-pressure phase Si(*tP*12/*tP*20) can be observed upon thermal 200 °C ex situ annealing of the Si(c*I*16)/Si(*hR*24) mixture made by indentation, and this is impossible to replicate in a DAC [47]. In addition to the differences above, the indentation differs from DACs in the geometry of the transformed material. While the material formed via DAC is limited in its geometry to a solid monolithic material, large arrays of indentations can be made that result in a thin film of metastable Si on a standard silicon wafer. Thus, indentation is readily more exploitable for technological applications, if the extensive defects are suppressed or eliminated.

#### 3.1.3. Laser-Induced Method

Compared with the diamond anvil and indentation methods, the ultra-short laser pulse-induced phase is a more favorable route, offering faster heating/cooling and pressure/decompressure rate, which is quite effective for the generation of submicron silicon metastable grains. Most of the previous research focused on the direct effect of laser on silicon wafers to obtain other phases under different irradiation conditions, such as *a*-Si, Si(*hR*24) and Si(c*I*16) [70]. Smith and his coworkers did notable work on the mechanism of laser-induced phase transition. In 2011, they reported that the dopant precursor does not directly impact the formation of high-pressure phases of silicon, despite its influence on the microstructure [70]. Additionally, they revealed the light–material interaction during the texturing of silicon by correlating the formation of pressure-induced silicon polymorphs, fs-laser irradiation conditions, and the resulting morphology and microstructure [71]. On the basis of these observations, they proposed the mechanism of phase transformation; the re-solidification of molten silicon on a textured surface plays a central role in driving sub-surface pressure-induced phase transformations and the effect of dopant precursor is too little to consider. The thermal stability of silicon polymorphs is superior to that produced by nanoindentation, which is a result of the limitation of the *a*-Si matrix. 

In contrast, Rodichkina and others [24] observed photo-induced cubic-to-hexagonal polytype transition in silicon nanowires under laser irradiation at intensities above 10 kW·cm^−2^ and the red-shift of PL maximum peak with the increase in photoexcitation level. The stress distribution in two-dimensional nanowires is different from that in the bulk silicon, as described in Section 3.1; thus, the resulting phase is different, necessitating further investigation via finite element analysis.

Overall, it is promising to discover more new forms of silicon by cooperatively adjusting variables such as the working medium, laser power density, and the width of pulse. Importantly, to achieve control over the phase formation, it will be essential to fully understand the laser–sample interaction during such reaction processes.

#### 3.1.4. Metal-Assisted Chemical Etching Method

The metal-assisted chemical etching method (MACE) is a facile top-down technique for the fabrication of porous silicon, which can be used to prepare wurtzite Si nanowires. In MACE, the silicon substrate is coated with noble metal (Au, Ag) as a catalyst for etching of silicon, and then immersed in the etching solution consisting of etchant, oxidant, and dilutant to form Si nanowires. Raman spectra demonstrated the vertically aligned arrays of synthesized Si nanowires involve Si(*hP*4) and Si(*cF*8) nanowires with quantum confinement. Such silicon can also be used as antireflecting material due to its high absorption coefficient in the visible/near infrared region and special morphology. Shukla [23] determined an optimal deposition time of 20 s and etching time 30 min, and they believed that the difference in surface energy between hydrogenated silicon and silicon was the essence for the induced silicon phase transition. 

From Section 3.1.1. and Section 3.1.3, we know that shape can change the distribution of internal stress in a material. Therefore, can other new metastable phases appear if we etch the metastable phases or precursors with different morphologies?

#### 3.1.5. Shear-Drive Method

In 2014, Vincent [72] reported that the volume expansion from hardening hydrogen silsesquioxane (HSQ) to silica filled in Ge nanowires promotes the formation of hexagonal and cubic Ge heterojunctions (Figure 8a,b); then, they [73] performed the same operation on Si nanowires and again obtained a cubic/hexagonal heterostructure. More importantly, they found that the silicon nanowires with different orientations have different Schmid factors, and a higher Schmid factor leads to easier slipping (Schmid factor [1 1 1] > [1 1 0] > [1 1 3] > [1 0 0] for the < 5 5 2 > {1 1 5} shear band). However, only few atomic-scale Si(*hP*4) nanoribbons at the base of the fins rather than Si(*hP*4) and Si(*cF*8) heterojunctions could form during the oxide fill step in the FinFET structures (Figure 8c), indicating that shapes responded to the lateral outward stress differently.

#### 3.1.6. Ball Milling Method

Mechanochemical methods are technologies that use an intensive mechanical force (squeeze, friction, shear and impact) to purposely activate powders, changing the crystal structure and chemical activity of particles to a certain extent. On the base of Hertzian Impact Theory, Kulnitskiy [74,75] believed that collision is the major factor generating over 94% of the energy transferred, and high-temperature and high-pressure conditions (>8 GPa, >698 K) can be achieved during ball milling. Indeed, it is notable that various particles undergo distinctive processing and, therefore, many exotic phases, such as Si(c*I*16) and Si(*hP*4), can form. However, the fatal shortcoming of the ball milling method is that its final products are always polytype mixtures rich in defects, thereby lowering their integrability.

#### 3.1.7. Metal-Induced Crystallization Method (MIC)

The metal-induced crystallization method has emerged as a viable technique for forming new crystal structures as the result of the rearrangement of the Si covalent bond near the metal/silicon interface through depositing a metal on an *a*-Si film or directly implanting metal ions into the substrate. In MIC, metals, when in contact with Si, act as a “seed” for crystallization, lower the crystallization temperature, and shorten anneal times. 

In 1993, Zhang et al. [76] first discovered the catalytic effect of a nickel layer on *a*-Si crystallization. Later, people successively discovered that other metals can also promote the crystallization of silicon. Mohiddon and Krishna’s groups at the University of Hyderabad have conducted extensive research on metal layer-induced amorphous silicon crystallization (chromium [77], nickel [78,79]) for the past few years. They studied the thermal behavior such as diffusion and reaction under different annealing temperatures, and then put forward the crystallization mechanism of *a*-Si: (1) interdiffusion between metal and silicon; (2) reaction of metal and silicon to form silicide; (3) silicon nanocrystalline grows on the silicide seeds. Furthermore, they found that the silicon nanocrystals in the silicide matrix exhibited a wide band gap, high resistivity, high transmittance, and a lower refractive index, making this material very feasible for optoelectronic and microwave applications.

The ion implantation method is a semiconductor doping technology developed in the 1960s and is the basis for obtaining *p*–*n* junctions. In a previous study, the hexagonal phase was observed in silicon upon irradiation with medium-energy ions, particularly for very high doses of phosphorus ions. Tan [80] obtained rod-like wurtzite silicon with As^+^ as dopants in 1981. The existence of a hexagonal phase in a diamond-cubic silicon wafer implanted at high dose rates, as well as in silicon mechanically deformed at temperatures between 500 and 600 °C, has been detected, which may be induced by a uniaxial compressive stress. Subsequently, many other metal ions such as Kr^+^ and Ga^+^ have been implanted into silicon or silica/silicon substrate to explore the mechanism and discover more new phenomena [81,82].

### 3.2. Bottom-Up Approach

#### 3.2.1. Wet Chemistry Method

Various liquid routes such as reduction of silicon halides with sodium napthalide [83], oxidation of metal silicide with halogens/NH_4_Br [84,85], and metathesis reaction between silicon halides and metal silicide [86] have been reported to generate Si nanoparticles in the diamond cubic structure. One might ask whether it is possible to synthesize other new phases via the wet chemistry method.

Ganguly [33] reported the synthesis of Si NPs crystallizing in the Si(c*I*16) via a colloidal route for the first time. The quasi-spherical nanoparticles with an average particle size of 5.0 ± 0.9 nm could remain stable during heating up to 650 °C. According to theoretical calculations for an Si(c*I*16) nanoparticle of diameter 4–8 nm, the gap should be closer to 1.37 eV. Unexpectedly, Si NPs prepared by colloidal routes show blue rather than red emission, which may be ascribed to N or O impurity defect sites. It is necessary to further explore how to remove the defects of N and O impurities to obtain pure phase Si(c*I*16) and achieve continuous tunability of wavelength by controlling the concentration of impurities.

In 2014, Nae-Man Park [87] used sodium methylsiliconate as a water-soluble Si precursor and potassium iodide (KI) as a gold (Au) etchant to produce Si(*hP*4) nanowires from an Au film or particles at low temperature and atmospheric pressure. In this sample liquid phase route, the raw materials were nontoxic and the nanocrystalline was composed of pure Si. However, there were no tests related to the performance of the material in this article.

From the available literature, we know that some liquid-phase methods can provide transient/long-term high-temperature and high-pressure conditions, such as ultrasonic deaeration, ultraviolet decomposition, and supercritical fluids. Conditions far from equilibrium may enable the formation of metastable phase, deserving further exploration and development. If batch controllable synthesis is possible, it will play an important role in industrial production.

#### 3.2.2. Chemical Vapor Deposition Method

The chemical vapor deposition method uses gaseous precursors to generate a solid ultrapure material through an atomic/intermolecular chemical reaction, possessing the ability to control the nanostructure (nanocrystal, nanofilm, nanowire, and nanocone) of the deposited material. Since the 1960s, CVD has been widely employed in semiconductor thin films (monocrystalline/polycrystalline/amorphous), as well as dielectric and diffusion barriers in integrated circuit. It is well known that the lowest energy state of silicon is Si(c*F*8). Thus, can other phases appear during the VLS/VSS mechanism growth process? This is usually interpreted as excessive energy input or surface stress induced by size effect. Actually, some surprising Si phases have grown in CVD through the use of different metal catalysts such as Au [88], Sn [89], Cu [90], Al [57], and In [91].

Palma [89] grew nanostructured silicon using a microwave/nano-susceptor technique with low substrate temperature. They discovered the formation of Si(c*I*16) in nanowires at low absorbed energy, whose ratio increased with microwave power but attenuated with gas pressure. Additionally, a high concentration of silicon in the tin droplets and the presence of plasma will lead to the growth of nano globules instead of nanowires.

Fontcuberta and Morral’s research team has kept ahead in the study of Si(*hP*4) nanowires synthesized by CVD. In 2007, they [88] first synthesized Si(*hP*4) nanowires on a gold-plated silicon substrate by CVD and explained the formation mechanism of Si(*hP*4) from the perspective of thermodynamics. When r is extremely small, surface energy is the dominant factor, and △G_f_(r) is proportional to the difference in surface energies, whereas, at relatively low values of r, the excess pressure resulting from the surface stress can induce the transformation from Si(*cF*8) to Si(*hP*4). Raman measurements [92] showed that the center of the Raman characteristic peaks of Si(*hP*4) were located at 506 cm^−1^ and 516 cm^−1^. It is worth noting that the critical value of r has not yet been determined, and the Si(*hP*4) can gradually evolve to Si(*cF*8) through the formation of dislocation. The control of diameter, thus, may be the key to the formation of silicon allotropes, involving many factors such as contact angle of the solid–liquid interface, temperature, and pressure/partial pressure [93,94,95]. The same year, they tried to prepare silicon nanowires with Cu as a catalyst via the vapor–solid–solid process and found that the structure of the wires changed continuously along the growth direction from diamond to wurtzite. The results will be important for the integration of silicon nanowires and Cu interconnects.

Silicon epitaxial growth technology usually requires unconventional templates. to meet the requirement of crystal integrity and high purity of metastable phases. In 2011, Bakkers [96] reported that Si(*hP*4) shells with thickness between 5 and 170 nm were grown on wurtzite GaP nanowires by MOCVD. In this method, the length of the twin superlattice (TSL) structure segment was controlled by changing the growth pressure of the Ga precursor to achieve precise control of the structure and properties of the silicon nanowires. After that, they proved the high stability of pure-phase Si(*hP*4) nanotubes at 9 GPa through a high-pressure Raman test, which provided the basis for the further study of its electrical, optical, and mechanical properties [97]. Recently, they grew direct-band gap hexagonal Ge and SiGe alloys using GaAs instead of GaP as core, and the measured result of sub-nanosecond, temperature-insensitive radiative recombination lifetime showed its excellent optical properties, similar to those of direct band gap group-III–V semiconductors [41]. The crystal structure of epitaxy growth can be highly determined by the substrate, making it possible to synthesis more novel forms of silicon.

Research progress related to metastable silicon prepared via CVD has mainly focused on three aspects: (1) growing perfect single-crystalline forms without defects and deep traps of catalyst; (2) reducing the deposition temperature to meet the growth of integrated circuits and flexible substrates [91]; (3) achieving *p*-type and *n*-type doping to obtain *p*–*n* junctions [5].

## 4. Challenges and Perspectives for the Field

### 4.1. The Key Troubles

Of course, the controlled synthesis of new silicon materials with tailored functionality is truly vital to satisfy future technological and societal needs; however, some processing and mechanism challenges are yet to be addressed for these emerging metastable structures. Next, we consider key challenges and opportunities in this field.

How can pure silicon phases be synthesized? Most of them coexist in a mixture of polymorphs or stacking faults, which hinders the characterization and application of material. Computational materials science has provided excellent predictions of performance; however, so far, few precise performances have been reported.

As detailed here, these metastable phases are dynamically stable but thermally unstable under atmospheric pressure, such that they may gradually turn into Si(*cF*8), limiting their application conditions and scope. Therefore, the solution of how to effectively improve the stability is crucial.

Last but not least, applying real-time, in site, noninvasive analysis to the growth of exotic silicon phase seems particularly significant. Advanced detection technologies can make it easier to explore the growth mechanism and to match between the parameters and final result. However, creating an effective approach which fits the growth conditions remains unsolved.

### 4.2. Research Trends

According to the international research trends, we can see that the preparation of functional pure phase metastable silicon is still a key point in the semiconductor field. Therefore, three emerging mainstream research directions deserve our attention.

On the theoretical front, computational materials science approaches for crystal structures and property prediction have been making headway. They enable determination of the atomistic mechanisms and transition pathways for these complicated phase transitions. Clearly, a combination of experimental and new computational methods will lead an accelerated pace of progress in the near future.

Carbon and germanium, as the adjacent elements in the same group of silicon, possess similar features such as crystal and electronic structures. Likewise, the synthetic methods, similar to C/Ge metastable allotropes, can also be applied to yield further silicon. Moreover, the interest in the class of group IV alloy has significantly increased in recent years due to the possibility of obtaining a direct band gap material and other special properties on a single silicon crystal.

Scaling these approaches up to technologically interesting levels is the only way. To achieve the goal of industrialization, not only the size, but also the defects and impurities in the material need to be precisely controlled, which is the first issue that the semiconductor industry needs to consider. Hence, a reduction in the number of defects and impurities will be the priority if the goal of pure silicon allotropes is to be successfully achieved. Future studies on high-quality metastable silicon grown via CVD would be helpful for commercial applications, because this method is more compatible and smaller in size compared to others, and there have already been surprising results.

## Figures and Tables

**Figure 1 materials-14-03964-f001:**
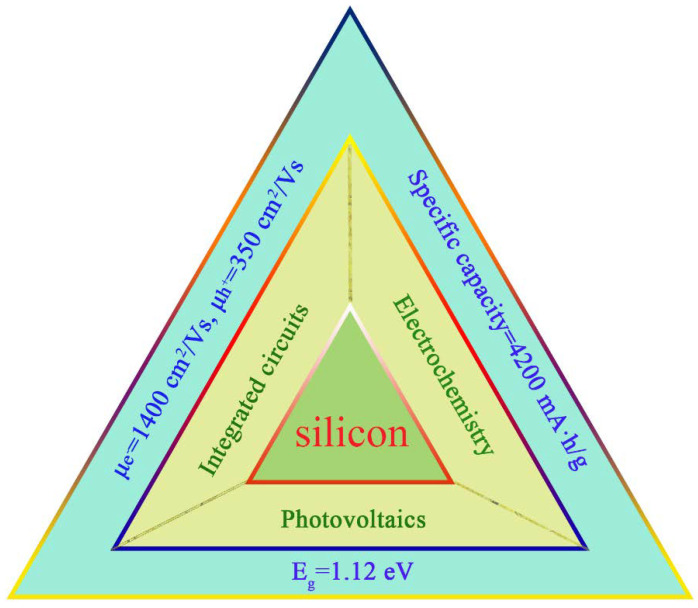
Properties and typical applications of semiconductor silicon of semiconductor silicon.

**Figure 3 materials-14-03964-f003:**
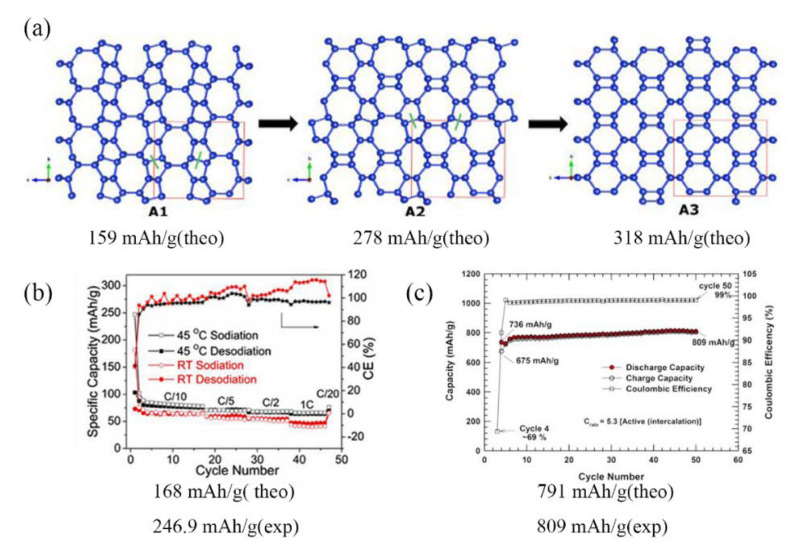
Specific capacity of clathrates of silicon. (**a**) Si(*oC*24). Inter-cages Si–Si breaking followed by full relaxation leads to the formation of two unreported allotropes A2 and A3. Green lines indicate the Si–Si bond breaking. Reprinted with permission from Ref. [12]. Copyright 2016, Electrochimica Acta. Capacity and coulombic efficiency of a Si(*cP*46) electrode during cycling at the 50th cycle: (**b**) Si(*cF*136). Adapted with permission from Ref. [13]. Copyright 2019, Journal of the Electrochemical Society. (**c**) Si(*cP*46). Adapted with permission from Ref. [14]. Copyright 2016, Journal of Materials Research.

**Figure 4 materials-14-03964-f004:**
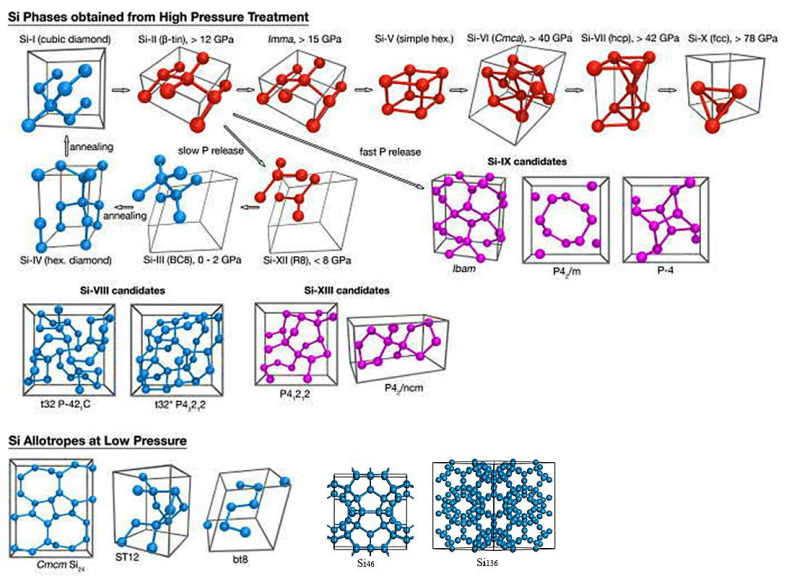
Schematic overview of as-prepared phases of silicon. Structures known to exist in ambient conditions are marked in blue. High-pressure phases are in red. Violet denotes structures proposed for the still elusive Si-IX and Si(*tP*12/*tP*20) phases. Adapted with permission from Ref. [7]. Copyright 2016, Applied Physics Reviews. The original quoted figure lacked Si_46_ and Si_136_, which were additionally drawn in this figure.

**Figure 5 materials-14-03964-f005:**
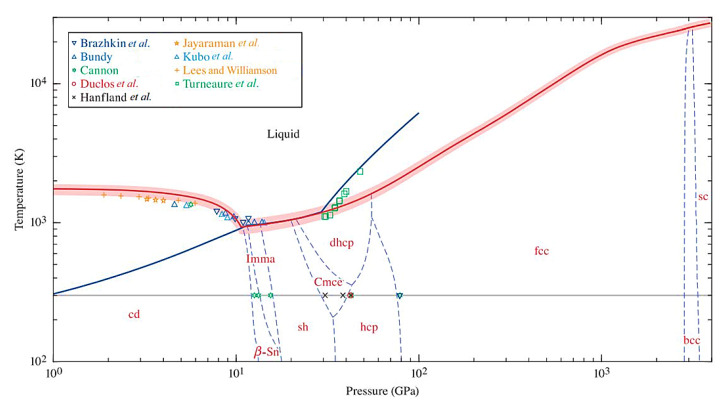
Pressure-temperature (P-T) phase diagram of silicon predicted using first-principles methodology. Here, the gray horizontal line represents the T = 300 K isotherm, whereas the solid blue line represents the principal shock Hugoniot. The discrete data points, which are also labeled in the legend, correspond to experimentally observed phase-transition points. Reprinted with permission from Ref. [15]. Copyright 2019, Physical Review B.

**Figure 6 materials-14-03964-f006:**
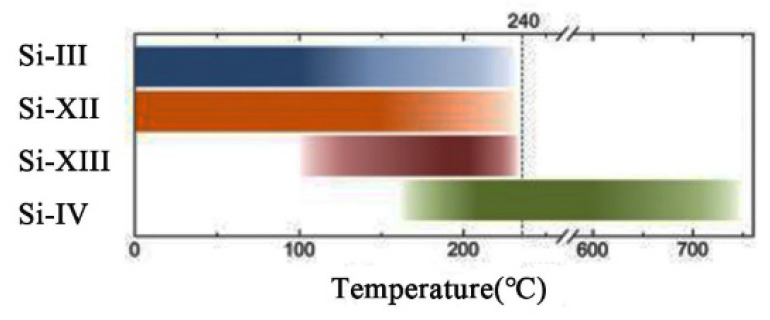
A plot showing the presence of the exotic Si phases after thermal annealing at varying temperatures. Fading colors represent temperatures at which the phase is gradually transforming. The temperature at which the Si(*cI*16) to Si(*hP*4) transformation begins is an estimate only. Adapted with permission from Ref. [22]. Copyright 2019, Journal of Applied Physics.

**Figure 7 materials-14-03964-f007:**
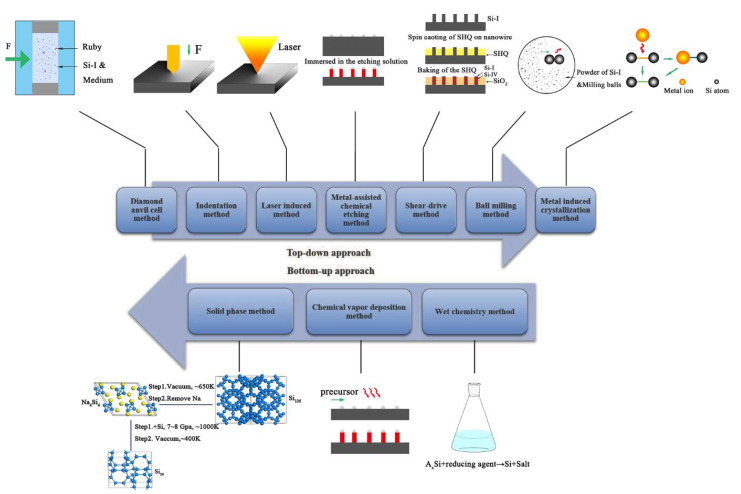
Synthetic methods of metastable silicon allotropes.

**Figure 8 materials-14-03964-f008:**
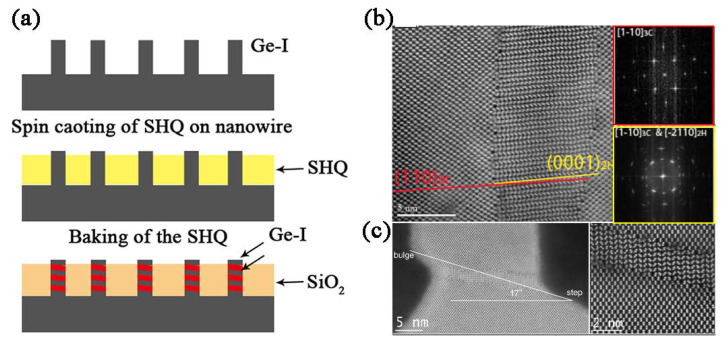
(**a**) Schematic of the fabrication procedure of Ge-I/Ge-IV heterojunctions. Microstructure of cubic/hexagonal heterojunction prepared by the shear-drive method. (**b**) High-resolution bright-field STEM image of the heterostructured Ge nanowires. Adapted with permission from Ref. [72]. Copyright 2014, Nano Lett. (**c**) Atomic scales nanoribbon with angle of 17°. Adapted with permission from Ref. [46]. Copyright 2015, Sci Rep.

**Table 1 materials-14-03964-t001:** Structural information of silicon allotropes [8,18,20,21,22,23,24,25,26].

Label	Space Group	Pearson Symbol	Structure	Main Raman Band (cm^−1^)	Relative Energy (eV/atom)
Si-I, DC-Si	*Fd* $\bar{3}$ *m*	*cF*8	Cubic	520	0
Si-XII, R8-Si	*R* $\bar{3}$	*hR*24	Rhombohedral	352	0.16
Si-III, BC8-Si	*Ia* $\bar{3}$	*cI*16	body-centered cubic	438	0.159
Si-IV, HD-Si	*P*6_3_/mmc	*hP*4	hexagonal diamond	515~517	0.011
Si-XIII	*P*4_2_/ncm *P*4_1_2_1_2	*tP12*/*tP*20	Tetragonal	475	0.045/0.041
Si-VIII, t32/t32*-Si	*P*$\bar{4}$2_1_*C* *P*4_3_2_1_2	*tP*32	Tetragonal	-	0.163/0.164
BT8-Si	*I*4_1_*/a*	*tI*16	Body-centered tetragonal	-	0.146
ST12-Si	*P*4_3_2_1_2	*tP*12	Simple tetragonal	-	0.139
Si_24_	*Cmcm*	*oC*24	Orthorhombic	-	0.09
Si_46_	*Pm* $\bar{3}$ *n*	*cP*46	Cubic	-	0.067
Si_136_	*Fd* $\bar{3}$ *m*	*cF*136	Face-centered cubic	487	0.052

“Raman band” displays the most prominent frequency for each phase. Note that Si-VIII has two candidates, t32(*P*$\bar{4}$2_1_*C*) and t32*(*P*4_3_2_1_2).

**Table 2 materials-14-03964-t002:** Experimental and theoretical band structure of Si(*hP*4).

Indirect Band Gap Width (eV)	Direct Transition at Γ Point (eV)	Ref.
0.92	1.53	Filippo Fabbri et al. (exp) [6]
0.85	1.40	Joannopoulos and Cohen (theo) [37]
0.99	-	Persson and Janzen (theo) [38]
0.92	-	Raffy et al. (theo) [39]
0.95	1.63	C. Rödl et al. (theo) [40]

## Data Availability

No new data were created and analyzed in this study.

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
