# Peer review of "A Review on Metastable Silicon Allotropes"

_materials, 2021, doi:10.3390/ma14143964_

Round 1

Reviewer 1 Report

materials-1208251                             

A review on metastable silicon allotropes

This manuscript by Fan et al. attempts to review the state of the art in synthesis, structure prediction, and properties of silicon allotropes. The topic is timely and of interest, but this manuscript has many problems. Although such a review article, if it provided more insight and perspective than available in already published recent reviews by others, could be a useful contribution, this manuscript does not meet the quality standards for publication in its current form. For the reasons summarized below, major revisions, including reorganization, improved writing, and expanded discussion, would be required before the paper can really be considered for publication. In fact, these issues need to be addressed before the content of the paper can be adequately evaluated.

1. There have been many review articles recently published covering the topic of the present review, some of which the authors cite. I do not see a compelling case presented in this manuscript for another review article, i.e. it is not clear what this additional review contributes that cannot be found in the previously published reviews. The authors could, for example, expand the discussion of how the properties are related to structure, and provide guidance on which kinds of structures are most promising for various applications.

2. The organization of the paper needs to be significantly improved. For example, the introduction is very long (6 pages!), and already the introduction is reviewing the literature in detail. I suggest to shorten the introduction to perhaps a paragraph or two, to motivate the review, and then organize the remaining discussion into appropriate sections.

3. Currently, the nomenclature used in the manuscript for the different allotrope structures is not consistent and somewhat confusing. The authors should adopt a common and consistent nomenclature for the all of the different crystalline structures of the various allotropes that are discussed. It is strongly recommended to follow the IUPAC nomenclature of elemental polymorphs which includes the Pearson symbol, e.g. Si136 would be Si(cF136), etc.

4. The font size in several of the figures, namely Figs. 3, 6, 8, 9, is far too small to be legible. Please revise the figures so that all text is clear and legible.

5. The figure caption for Figure 3 does not correctly describe the figure panes. Please revise.

6. Since this is a review article, it is very important that all work in this area is referenced appropriately in order to be comprehensive. There have been a number of reviews and highlights covering different forms of silicon published in recent years. The authors have referenced several of these but missed at least two that should also be cited, in particular in the discussion of clathrates and other low-density allotropes:

- M. Beekman, “New hopes for allotropes,” Mater. Today 18, 304 (2015).

- M. Beekman, K. Wei, and G. S. Nolas, “Clathrates and beyond: low-density allotropy in crystalline silicon,” Appl. Phys. Rev. 3, 040804 (2016).

Some early theoretical work on silicon allotropes has also been neglected:

- M. O’Keeffe, G. B. Adams and O. F. Sankey, Philos. Mag. Lett. 78, 21 (1998).

- G. B. Adams, M. O’Keefe, A. A. Demkov, O. F. Sankey, and Y.-M. Huang, Phys. Rev. B 49, 8048 (1994).

The authors should revisit the literature and make sure that their review is comprehensive.

7. “Cubic diamond structure is the thermodynamic ground state of silicon…” Please clarify that this is true at room temperature and atmospheric pressure.

8. There are factual errors in the paper. For example:

Line 299: It is asserted that Si136 has “ten-membered rings.” I fail to see this. There are only 5- and 6-membered rings.

Line 319: It is asserted that the structure of Si46 only has Si20 cages. This is not correct.

9. The authors could mention that Si136 is stable to nearly 600 deg C (even higher thermal stability than Si24), as reported in:

- Beekman and G.S. Nolas, “Synthesis and thermal conductivity of type II silicon clathrates,” Physica B 383, 111 (2006).

10. The claims reported in Refs. 19 and 20 about synthesis of Si46 are likely not correct. The samples reported in Refs. 19 and 20 were poorly characterized and no convincing evidence was provided that Si46 was prepared in any significant fraction. The authors should be very careful about promoting unsubstantiated claims in their review article.

11. Please carefully check every reference to ensure they are correctly cited in the manuscript. For example, Ref. 26 is cited on line 121 but this reference does not seem appropriate.

12. The written English needs to be greatly improved. Grammatical errors and typos are found throughout the manuscript. Please consider having the manuscript proofread by a native speaker. Examples include (but are not limited to):

Abstract: “indirect-gap band”

Abstract: “band gaps allowed efficient”

Line 40: “the earth-abundant storage and mature industrial processing methods attributes to Si is extremely sought after”

Line 121: I believe the authors mean to write “thermal decomposition” and not “thermal decompress.”

Line 124: “allomorphs will brings’’ should be “allotropes” or “polymorphs?” And, “brings” should be “bring.”

Throughout: Words are capitalized that should not be capitalized.

I emphasize that the above are just a few examples found on the first few pages. There are many more corrections that need to be made (too many to list).

Author Response

Zhejiang University

                                                                                                         24 May 2021

Dear Editor and reviewer,

We are submitting our revised manuscript entitled ‘A review on metastable silicon allotropes’ for publication in “Materials”.

Thanks for the reviewer’s valuable comments. The comments of the reviewer were highly insightful and enabled us to improve the quality of our manuscript. We have responded to reviewer’s comments points to points, and have made the corresponding changes in the revised manuscript. In the following page is our response to the comments of the reviewers.

Revisions in the text are highlighted for additions and changes.

We look forward to hear from you at your earliest convenience.

Best regards,

Dongsheng, Li

Reviewer #1:

Comment #1: There have been many review articles recently published covering the topic of the present review, some of which the authors cite. I do not see a compelling case presented in this manuscript for another review article, i.e. it is not clear what this additional review contributes that cannot be found in the previously published reviews. The authors could, for example, expand the discussion of how the properties are related to structure, and provide guidance on which kinds of structures are most promising for various applications.

Response: Thank you for the comment. I re-read these recently published review articles mentioned by the reviewer. I pay more attention to the links between these reviews and the results mentioned in them. The discussion of how the properties are related to structure, and provide guidance on which kinds of structures are most promising for various applications were added in revised manuscript. They are listed below this letter:

  1. Line 127: The narrow-gap semiconductor of Si(Ia) is a highly desirable feature for multiple exciton generation based solar cells and the simulated maximum energy conversion efficiency can reach 51.6% due to effective usage of high-energy photon.
  2. Line 155: Si(R) has a unique form among these metastable silicon phases that it contains both five- and six-membered rings in rhombohedral unit cell, the presence of which may account for the special electronic abilities of the material for high carrier mobility and electrically activated at room temperature. Taking advantage of the high carrier mobility and a ~50% lower reflectance, Mannepalli et al.[11] make a silicon solar cell with a few hundred micrometers Si(R) ordered dotted array on the top layer, which show ~10 times improvement in the photocurrent density
  3. Line 222: Such open-cage allotropes serve as diffusion channels and reservoir for ions, with suitable shape and size.
  4. Line 225: What`s more, the theoretically predicted band gap is wider than the diamond cubic silicon, illustrating the more overlap of absorption with the solar spectrum.

Comment #2: The organization of the paper needs to be significantly improved. For example, the introduction is very long (6 pages!), and already the introduction is reviewing the literature in detail. I suggest to shorten the introduction to perhaps a paragraph or two, to motivate the review, and then organize the remaining discussion into appropriate sections.

Response: Thanks for the comment. I have condensed the introduction and merged the language of the excessive summary into the second chapter. When analyzing the relationship between structure and performance, I described and generalized the current research progress in detail.

Comment #3: Currently, the nomenclature used in the manuscript for the different allotrope structures is not consistent and somewhat confusing. The authors should adopt a common and consistent nomenclature for the all of the different crystalline structures of the various allotropes that are discussed. It is strongly recommended to follow the IUPAC nomenclature of elemental polymorphs which includes the Pearson symbol, e.g. Si136 would be Si(cF136), etc

Response: Thanks for the comment. I have obtained the Pearson symbol of each phase in this review by consulting the literature, and updated the IUPAC nomenclature in the article.

Comment #4: The font size in several of the figures, namely Figs. 3, 6, 8, 9, is far too small to be legible. Please revise the figures so that all text is clear and legible.

ResponseThanks for the comment. It can be seen from the following pictures that I have optimized the clarity of the pictures and adjusted the order of the pictures.

  1. Merged original figure 3 and figure 2.
  2. Removed original figure 8.

Revised figure 2

Revised figure 5

Revised figure 7

Comment #5: The figure caption for Figure 3 does not correctly describe the figure panes. Please revise.

Response: Thanks for the comment. The figure caption after revised can be found in line 774.

Comment #6: Since this is a review article, it is very important that all work in this area is referenced appropriately in order to be comprehensive. There have been a number of reviews and highlights covering different forms of silicon published in recent years. The authors have referenced several of these but missed at least two that should also be cited, in particular in the discussion of clathrates and other low-density allotropes:

- M. Beekman, “New hopes for allotropes,” Mater. Today 18, 304 (2015).

- M. Beekman, K. Wei, and G. S. Nolas, “Clathrates and beyond: low-density allotropy in crystalline silicon,” Appl. Phys. Rev. 3, 040804 (2016).

Some early theoretical work on silicon allotropes has also been neglected:

- M. O’Keeffe, G. B. Adams and O. F. Sankey, Philos. Mag. Lett. 78, 21 (1998).

- G. B. Adams, M. O’Keefe, A. A. Demkov, O. F. Sankey, and Y.-M. Huang, Phys. Rev. B 49, 8048 (1994).

Response: Thanks for the comment. We sincerely appreciate the valuable comments. We have checked the literatures carefully and added more references in the revised manuscript.

Line 221: The important progresses on low density Si clathrates have been well discussed in the reviews entitled “New hopes for allotropes”, “Exotic forms of silicon” and “Clathrates and beyond: low-density allotropy in crystalline silicon”. Such open-cage allotropes serve as diffusion channels and reservoir for ions, with suitable shape and size. What`s more, the theoretically predicted band gap is wider than the diamond cubic silicon, illustrating the more overlap of absorption with the solar spectrum. Here, we supplement some research work on their basis for nearly five years.

Line 230: M. O’Keeffe found the type II structure have the lowest energy in all clathrates using empirical potentials.

  1. Beekman. New hopes for allotropes. Materials Today, 2015, 18(6): 304-305.
  2. C. Taylor. Exotic forms of silicon. Physics Today, 2016, 69(12): 34-39.
  3. W. Matt Beekman1, and George S. Nolas2,*. Clathrates and beyond: low-density allotropy in crystalline silicon. AIP Publishing, 2016.
  4. B. Adams, M. O'Keeffe, A.A. Demkov, O.F. Sankey, and Y.M. Huang. Wide-band-gap Si in open fourfold-coordinated clathrate structures. Phys Rev B Condens Matter, 1994, 49(12): 8048-8053.
  5. O'Keeffe, G.B. Adams and O.F. Sankey. Duals of Frank-Kasper structures as C, Si and Ge clathrates: Energetics and structure. Philosophical Magazine Letters, 1998, 78(1): 21-28.

Comment #7: “Cubic diamond structure is the thermodynamic ground state of silicon…” Please clarify that this is true at room temperature and atmospheric pressure.

Response: Thanks for the comment. I have corrected this wrong expression.

Line 68: From it we can know cubic diamond structure is the thermodynamic ground state of silicon, the others are metastable or unstable phases at room temperature and atmospheric pressure condition…

Comment #8: There are factual errors in the paper. For example:

Line 299: It is asserted that Si136 has “ten-membered rings.” I fail to see this. There are only 5- and 6-membered rings.

Line 319: It is asserted that the structure of Si46 only has Si20 cages. This is not correct.

Response: Thanks for the comment. I carefully consulted the literature on Si(cF136) and Si(cP46), and verified the structures through modeling, and got the following conclusions:

  1. There are only 5- and 6-membered rings in Si(cF136).

Line 233: The cage-like structure Si(cF136) (also called type-II silicon clathrate) possesses 136 Si atoms per unit cell (cF136) in sp3-like arrangement, which consists of five- and six-membered rings.

Line 258: Type I silicon clathrate Si46(cP46), which consists of Si with a regular arrangement of 20-atom and 24-atom cages connected together through 5-atom pentagonal rings, has a simple cubic structure (a=10.335 Å) with 46 Si atoms per unit cell.

Comment #9: The authors could mention that Si136 is stable to nearly 600 deg C (even higher thermal stability than Si24), as reported in:

- Beekman and G.S. Nolas, “Synthesis and thermal conductivity of type II silicon clathrates,” Physica B 383, 111 (2006).

Response: Thanks for the comment. I have added the thermal stability of Si(cF136) to make the article more comprehensive.

Line 245: It is worth mentioning that Si(cF136) can remain stable to nearly 600 °C (even higher thermal stability than Si(oC24)), which meets its further applications in PV and batteries.

Comment #10: The claims reported in Refs. 19 and 20 about synthesis of Si46 are likely not correct. The samples reported in Refs. 19 and 20 were poorly characterized and no convincing evidence was provided that Si46 was prepared in any significant fraction. The authors should be very careful about promoting unsubstantiated claims in their review article.

Response: Thanks for the comment. The XRD peak positions of the product in Reference 26 are consistent with the calculation result and JCPDS(No.01-089-5534), and the extra peak may be by-products due to incomplete purification. This shows to a certain extent that Si46 is indeed synthesized, but the purity is not high enough. We quote this reference here only to make a simple comparison, and show that the low-density cage structure is suitable for battery electrodes without too much consideration of purity and fraction.

In consideration of the comment, I have made a supplementary explanation that the electrode is a mixture instead of Si(cP46). (Line 262: The capacity of the Half-cells made from the mixture of Si(cP46) and by-products undergoes…).

Comment #11: Please carefully check every reference to ensure they are correctly cited in the manuscript. For example, Ref. 26 is cited on line 121 but this reference does not seem appropriate.

Response: Thanks for the comment. I have removed this inappropriate reference.

Line 84: Some low-density clathrates, such as Si136, need to be prepared by thermal decomposition the Zintl phase.

Comment #12: The written English needs to be greatly improved. Grammatical errors and typos are found throughout the manuscript. Please consider having the manuscript proofread by a native speaker. Examples include (but are not limited to):

Abstract: “indirect-gap band”

Abstract: “band gaps allowed efficient”

Line 40: “the earth-abundant storage and mature industrial processing methods attributes to Si is extremely sought after”

Line 121: I believe the authors mean to write “thermal decomposition” and not “thermal decompress.”

Line 124: “allomorphs will brings’’ should be “allotropes” or “polymorphs?” And, “brings” should be “bring.”

Throughout: Words are capitalized that should not be capitalized.

Response: Thanks for the comment. We tried our best to improve the manuscript and made some changes, but these changes will not influence the content and framework of the paper. And we hope the correction will meet with approval.

Line 10: However, its gap band allowed efficient deprives Si further developing.

Line 39: Although it is easy to realize photoelectric interconnection through III-V semiconductors, if silicon has the desirable properties, it will be the grail relying on the earth-abundant storage and mature industrial processing methods.

Line 83: Low-density clathrates, such as Si(cF136), need to be prepared by thermal decomposition the Zintl phase.

Line 87: It is thus highly believed that the research on silicon allomorphs will bring broad prospects to silicon-based materials.

Reviewer #2:

Comment: The manuscript is very interesting. Very good selection of literature and silicon allotropes shown. However, the author's weakness is to keep the right description in the right grammatical form. The manuscript contains many syntax errors. The illustrations look like they were cut from another publication and are of very low quality. Additionally, the author does not provide their original reference location.

Response: Thanks for the comment. I am sorry for the grammatical errors in this article, and I have corrected them. And I reconstruct the picture with high-definition images and give their original reference locations (as shown figure and table captions from page 16 to page 20). Checked the language expression in the entire manuscript and made the following corrections:

  1. Line 460: Various liquid routes such as reduction of silicon halides with sodium naphthalide/Na, oxidation of metal silicides with halogens/NH4Br, and comproportionation reaction between silicon halides and metal silicides have been reported to generate Si NPs in the diamond cubic structure.

Line 442: Various liquid routes such as reduction of silicon halides with sodium napthalide, oxidation of metal silicide with halogens/NH4Br, and metathesis reaction between silicon halides and metal silicide have been reported to generate Si NPs in the diamond cubic structure.

  1. Line 793, 831: Blue dott lines connect each phase from evolutionary algorithms and Solid-state thermodynamic calculations.

Line 787, 821: Blue dotted lines connect each phase from evolutionary algorithms and Solid-state thermodynamic calculations.

  1. Table 1: body-centred cubic, Orthohombic

Table 1: body-centered cubic, Orthorhombic

  1. Line 384: In contrast, S.P.Rodichkina and others…

Line 366: In contrast, S. P. Rodichkina and others…

  1. Line 802, 846: Few atomic scale nanoribbon with angle of 17°.

Line 795, 834: Atomic scales nanoribbon with angle of 17°.

Reviewer #3:

Comment #1:Line 69-72: Figure 3. Outstanding performances of silicon allotropes in solar cells...

Response: Thanks for the comment. I merged Figure 2 and Figure 3, and corrected the title to “Fig.2 Outstanding performances of silicon allotropes in solar cells. (a) Comparison between CL spectra of 2H-Si nanowires and standard cubic silicon. Si nanowires have two light-emitting peaks at about 1.5 eV and 0.8 eV. (b) Calculated power conversion efficiencies of single stage Si(Ia) QDs solar cells at solar spectra AM1.5 G. (c) Reflectance spectra of the standard Si wafer, pristine n-p-p ++ solar cell wafer, the indented cell before and after annealing. The inset shows the optical image of indents made on cell and schematic shows the enhanced light absorption by the indented area. The symbols ε1 and ε2 indicate the dielectric functions of the Si(R) and Si(cF8), respectively. (d) The J-V characteristics of pristine cell(black) and indented cell(red)” (page 16).

Comment #2: Line 118,119: In addition to these phases attained from high pressure treatment, the experimental observation of ST12 and bt8 induced by ultrashort laser pulses is reported.

Response: Thanks for the comment. I have removed gray selection and you can see it from line 81.

Comment #3: Line 188,189: “Back in 1964, Drickamer and Minomura accidently found that when the pressure was as high as 200 KPa…

Response: Thanks for the comment. I’m sorry for the wrong author’s order and pressure, and I have corrected them in time. Line 113: Back in 1962, Minomura and Drickamer accidently found that when the pressure was as high as 200 kbar at room temperature, the resistivity of silicon became 10-5 ohm/cm, expressed as metallic.

Comment #4: Line 241,242: Comparing with other metastable phase, Si-I shows…

Response: Thanks for the comment. For this question, I use the Pearson symbol to distinguish the two strucures. Line 169: Comparing with other metastable phases, Si(hP4) shows relatively higher thermal stability and upon further annealing to 500 °C or even higher temperature Si(hP4) transforms back to the stable Si(cF8).

Comment #5: Line 488,489: …1. Interdiffusion between mental and silicon; …

Response: Thanks for the comment. I corrected the spelling mistakes and checked the full text to prevent similar mistakes. Line 426: Interdiffusion between metal and silicon; …

Comment #6: The same citation appears in 688 and 758 lines.

Response: Thanks for the comment. The correct year should be 2016 and I made the correction.

Comment #7: The year of the cited document is wrong in line 787.

Response: Thanks for the comment. The correct year should be 1981 and I made the correction (line 719).

Comment #8: The link of [89] is wrong.

Response: Thanks for the comment. I revised citation format of references. Line 734: F. Palma, E. Cattaruzza and P. Riello. Growth of nanostructured silicon by microwave/nano-susceptors technique with low substrate temperature. Materials Science in Semiconductor Processing, 2019, 100: 22-28.

Modifications to the manuscript

Pages 2:

I changed “Overall, the particular arrangements adopted by silicon atoms in each allotrope have a profound impact on the electrical, optical, thermal, and mechanical properties of silicon” to “Overall, atomic arrangement is closely related to material`s properties”.

Pages 4:

Line 192: (R) has a unique form among these metastable silicon phases that it contains both five- and six-membered rings in rhombohedral unit cell, the presence of which may account for the special electronic abilities of the material for high carrier mobility and electrically activated at room temperature.

Line 155: Si(R) has a unique form among these three metastable silicon phases that it contains five-membered rings in rhombohedral unit cell, the presence of which may account for the special electronic abilities of the material for high carrier mobility and electrically activated at room temperature.

Line 270: Such structure is suitable for membranes and battery anodes for molecular or ionic diffusion between reservoirs…

Line 251: Such structure is suitable for battery anodes for ion storage and diffusion…

Pages 6:

I deleted this sentence “In fact, theoretical calculations on the properties and structure of polymorphs have far exceeded the experimental preparation, but they can provide powerful guidance for synthesis in the laboratory”.

Pages 16-20:

  1. Figures 2 and 3 are both to express the excellent performance of heterogeneous silicon in photovoltaic applications, I put them together.
  2. I deleted the original image 8 because its meaning can be ignored and it is not appropriate to place it in this position.

Pages 2, 5, 6:

I changed the battery specific capacity unit from “mAh/g” to “mAh·g-1”.

Reviewer 2 Report

Dear Editor and Authors,

The manuscript is very interesting. Very good selection of literature and silicon allotropes shown. However, the author's weakness is to keep the right description in the right grammatical form. The manuscript contains many syntax errors. The illustrations look like they were cut from another publication and are of very low quality. Additionally, the author does not provide their original reference location.

Author Response

Dear Editor and reviewer,

We are submitting our revised manuscript entitled ‘A review on metastable silicon allotropes’ for publication in “Materials”.

Thanks for the reviewer’s valuable comments. The comments of the reviewer were highly insightful and enabled us to improve the quality of our manuscript. We have responded to reviewer’s comments points to points, and have made the corresponding changes in the revised manuscript. In the following page is our response to the comments of the reviewers.

Revisions in the text are highlighted for additions and changes.

We look forward to hear from you at your earliest convenience.

Best regards,

Dongsheng, Li

Comment: The manuscript is very interesting. Very good selection of literature and silicon allotropes shown. However, the author's weakness is to keep the right description in the right grammatical form. The manuscript contains many syntax errors. The illustrations look like they were cut from another publication and are of very low quality. Additionally, the author does not provide their original reference location.

Response: Thanks for the comment. I am sorry for the grammatical errors in this article, and I have corrected them. And I reconstruct the picture with high-definition images and give their original reference locations (as shown figure and table captions from page 16 to page 20). Checked the language expression in the entire manuscript and made the following corrections:

  1. Line 460: Various liquid routes such as reduction of silicon halides with sodium naphthalide/Na, oxidation of metal silicides with halogens/NH4Br, and comproportionation reaction between silicon halides and metal silicides have been reported to generate Si NPs in the diamond cubic structure.

Line 442: Various liquid routes such as reduction of silicon halides with sodium napthalide, oxidation of metal silicide with halogens/NH4Br, and metathesis reaction between silicon halides and metal silicide have been reported to generate Si NPs in the diamond cubic structure.

  1. Line 793, 831: Blue dott lines connect each phase from evolutionary algorithms and Solid-state thermodynamic calculations.

Line 787, 821: Blue dotted lines connect each phase from evolutionary algorithms and Solid-state thermodynamic calculations.

  1. Table 1: body-centred cubic, Orthohombic

Table 1: body-centered cubic, Orthorhombic

  1. Line 384: In contrast, S.P.Rodichkina and others…

Line 366: In contrast, S. P. Rodichkina and others…

  1. Line 802, 846: Few atomic scale nanoribbon with angle of 17°.

Line 795, 834: Atomic scales nanoribbon with angle of 17°.

Reviewer 3 Report

A very good review, the need for which is long overdue. It is well structured and can be published after correcting small remarks. There are two main remarks: 1. low resolution of figures 2. in table 1, the column with the coordination number should be removed, since it is everywhere equal to 4, then this should be added in the title of the table. Instead of this column, it is worth giving the values of the density of these phases (absolute, or in % of Si-I density). Minor notes are shown in the file.

Author Response

Dear Editor and reviewer,

We are submitting our revised manuscript entitled ‘A review on metastable silicon allotropes’ for publication in “Materials”.

Thanks for the reviewer’s valuable comments. The comments of the reviewer were highly insightful and enabled us to improve the quality of our manuscript. We have responded to reviewer’s comments points to points, and have made the corresponding changes in the revised manuscript. In the following page is our response to the comments of the reviewers.

Revisions in the text are highlighted for additions and changes.

We look forward to hear from you at your earliest convenience.

Best regards,

Dongsheng, Li

Comment #1:Line 69-72: Figure 3. Outstanding performances of silicon allotropes in solar cells...

Response: Thanks for the comment. I merged Figure 2 and Figure 3, and corrected the title to “Fig.2 Outstanding performances of silicon allotropes in solar cells. (a) Comparison between CL spectra of 2H-Si nanowires and standard cubic silicon. Si nanowires have two light-emitting peaks at about 1.5 eV and 0.8 eV. (b) Calculated power conversion efficiencies of single stage Si(Ia) QDs solar cells at solar spectra AM1.5 G. (c) Reflectance spectra of the standard Si wafer, pristine n-p-p ++ solar cell wafer, the indented cell before and after annealing. The inset shows the optical image of indents made on cell and schematic shows the enhanced light absorption by the indented area. The symbols ε1 and ε2 indicate the dielectric functions of the Si(R) and Si(cF8), respectively. (d) The J-V characteristics of pristine cell(black) and indented cell(red)” (page 16).

Comment #2: Line 118,119: In addition to these phases attained from high pressure treatment, the experimental observation of ST12 and bt8 induced by ultrashort laser pulses is reported.

Response: Thanks for the comment. I have removed gray selection and you can see it from line 81.

Comment #3: Line 188,189: “Back in 1964, Drickamer and Minomura accidently found that when the pressure was as high as 200 KPa…

Response: Thanks for the comment. I’m sorry for the wrong author’s order and pressure, and I have corrected them in time. Line 113: Back in 1962, Minomura and Drickamer accidently found that when the pressure was as high as 200 kbar at room temperature, the resistivity of silicon became 10-5 ohm/cm, expressed as metallic.

Comment #4: Line 241,242: Comparing with other metastable phase, Si-I shows…

Response: Thanks for the comment. For this question, I use the Pearson symbol to distinguish the two strucures. Line 169: Comparing with other metastable phases, Si(hP4) shows relatively higher thermal stability and upon further annealing to 500 °C or even higher temperature Si(hP4) transforms back to the stable Si(cF8).

Comment #5: Line 488,489: …1. Interdiffusion between mental and silicon; …

Response: Thanks for the comment. I corrected the spelling mistakes and checked the full text to prevent similar mistakes. Line 426: Interdiffusion between metal and silicon; …

Comment #6: The same citation appears in 688 and 758 lines.

Response: Thanks for the comment. The correct year should be 2016 and I made the correction.

Comment #7: The year of the cited document is wrong in line 787.

Response: Thanks for the comment. The correct year should be 1981 and I made the correction (line 719).

Comment #8: The link of [89] is wrong.

Response: Thanks for the comment. I revised citation format of references. Line 734: F. Palma, E. Cattaruzza and P. Riello. Growth of nanostructured silicon by microwave/nano-susceptors technique with low substrate temperature. Materials Science in Semiconductor Processing, 2019, 100: 22-28.

Round 2

Reviewer 1 Report

The authors have significantly improved the manuscript and addressed most of the concerns of the reviewers. The written English could still be substantially improved in many places. There are also some errors in some of the Pearson symbols, as well as in the references. The authors should fix these errors before the paper is published.

Author Response

Response letter

Zhejiang University

                                                                                                         7 July 2021

Dear Editor and reviewer,

We are submitting our revised manuscript entitled ‘A review on metastable silicon allotropes’ for publication in “Materials”.

Thanks for the reviewer’s valuable comments. The comments of the reviewer were highly insightful and enabled us to improve the quality of our manuscript. We have responded to reviewer’s comments points to points, and have made the corresponding changes in the revised manuscript. In the following page is our response to the comments of the reviewers.

Revisions in the text are highlighted for additions and changes.

We look forward to hear from you at your earliest convenience.

Best regards,

Dongsheng, Li

Comment: The written English could still be substantially improved in many places. There are also some errors in some of the Pearson symbols, as well as in the references.

Response: Thank you for the comment. We carefully checked these errors and corrected them in the revised manuscript. They are listed below this letter:

  1. The Pearson symbol of Si-III and Si-XIII are corrected to cI16 and tP12 in table 1.
  2. Fixed the subscript error in P63/mmc in table 1.
  3. Fixed the space group of BT8-Si and ST12-Si in table 1.
  4. The special symbols are set in italics in table 1.
  5. The text is synchronized with the same modification as above.
  6. Checked and improved the format in the references.

Line 570: S. Barth, M.S. Seifner and S. Maldonado. Metastable Group IV Allotropes and Solid Solutions: Nanoparticles and Nanowires. Chemistry of Materials, 2020, 32: 2703-2741.

Line 667: M. Beekman, K. Wei and G.S. Nolas. Clathrates and beyond: low-density allotropy in crystalline silicon. AIP Publishing, 2016, 3: 040804.

Line 685: T. Kume, F. Ohashi and S. Nonomura. Group IV clathrates for photovoltaic applications. Japanese Journal of Applied Physics, 2017, 56: 05DA05.

Line 695: A. Courac, Y.L. Godec, C. Renero-Lecuna, H. Moutaabbid, R. Kumar, C. Coelho-Diogo, C. Gervais, and D. Portehault. High-Pressure Melting Curve of Zintl Sodium Silicide Na4Si4 by In Situ Electrical Measurements. Inorg. Chem., 2019, 58(16): 10822–10828.

  1. We tried our best to improve the manuscript and made some changes, but these changes will not influence the content and framework of the paper. And we hope the correction will meet with approval.

Line 9:

Abstract: Silicon…

Abstract: Diamond cubic silicon…

Line 14:

However, its gap band allowed efficient deprives Si further developing.

However, it is a semiconductor with an indirect band gap, depriving its further development.

Line 14:

…or even optoelectronics

…and optoelectronics

Line 16:

…band gaps allowed efficient photoelectric conversion.

…band gaps allowed efficient for photoelectric conversion.

Line 25:

As the leading semiconductor, silicon lies at the heart of integrated circuits, photovoltaic and electrochemistry industry thanks to its distinct properties illustrated in figure 1[1, 2], providing an efficient and convenient lifestyle for humans.

Diamond cubic silicon lies at the heart of integrated circuits, photovoltaic and electrochemistry industry thanks to its distinct properties illustrated in figure 1[1, 2], providing a convenient lifestyle for humans.

Line 36:

Therefore, researchers have proposed a road…

Therefore, researchers proposed a road…

Line 43:

Silicon allotropes is a burgeoning branch, which exhibits…

Silicon allotrope is a burgeoning branch, which may exhibit…

Line 63:

…however, the experimental data are higher than the theoretical capacity…

…however, the experimental data are lagger than the theoretical capacity…

Line 69:

From it we can know cubic diamond structure is the thermodynamic ground state of silicon, the others are metastable or unstable phases at room temperature and atmospheric pressure condition, and the silicon crystal undergoes…

From it we can know cubic diamond structure is the thermodynamic ground state of silicon at room temperature and atmospheric pressure condition, and the crystal undergoes…

Line 76:

…they can exist stably only under…

…they can only exist stably under…

Line 89:

…the silicon allotropes have many types and varying properties…

…the silicon allotropes have many types and various properties…

Line 89:

This is followed by a discuss about challenges and perspectives for the field, before final concluding remarks are drawn.

Finally, the key problems and the developmental prospects are discussed at the end.

Line 102:

Si-III, Si-IV, and Si-XII, Si-VIII, BT8-Si, ST12-Si, Si24, Si46, Si136 in figure 5.

Si-III, Si-IV, and Si-XII, Si-VIII, Si-IX, BT8-Si, ST12-Si, Si24, Si46, Si136 in figure 5.

Line 113:

Back in 1962, Minomura and Drickamer [20] accidently found that when the pressure was as high as 200 kbar at room temperature, the resistivity of silicon became 10-5 ohm/cm, expressed as metallic.

Back in 1962, Minomura and Drickamer [20] accidently found that when pressure up to 200 kbar was applied at room temperature, the resistivity of silicon dropped to 10-5 ohm/cm, expressed as metallic.

Line 129:

or

and

Line 122:

The Raman peaks of Si(cI16) is around 160, 380, and 438 cm-1.

Line 163:

For the above three…

For the above three allotropes…

Line 197:

…how quickly it has been quenched to …

…how quickly it is quenched to…

Line 234:

The cage-like structure Si(cF136) (also called type-II silicon clathrate) possesses 136 Si atoms per unit cell (cF136) in sp3-like arrangement, which consists of five- and six-membered rings

The cage-like structure Si(cF136) possesses 136 Si atoms per unit cell in sp3-like arrangement, which consists of five- and six-membered rings.

Line 291:

…but are not suitable for…

…but not suitable for…

Line 353:

α-Si

a-Si

Line 380:

In MACE, silicon substrate is coated with noble metal (Au, Ag) as catalyst for etching of silicon to form Si nanowires…

In MACE, silicon substrate is coated with noble metal (Au, Ag) as catalyst for etching of silicon…

Line 417:

α-Si

a-Si

Line 419:

…lower the crystallization temperature and anneal times.

…lower the crystallization temperature and shorter anneal times.

Line 420:

α-Si

a-Si

Line 422:

Md. A. Mohiddon and M. G. Krishna's group at the University of Hyderabad has conducted…

Md. A. Mohiddon and M. G. Krishna's groups at the University of Hyderabad have conducted…

Line 429:

…large band gap…

…wide band gap…

Line 457:

…at low temperature under atmospheric pressure.

…at low temperature and atmospheric pressure.

Line 463:

They may enable…

Conditions far from equilibrium may enable…

Line 469:

…including the ability…

…possessing the ability…

Line 524:

How to synthesize pure phase? Most of them coexist in the form of polymorph or stacking faults…

How to synthesize pure silicon phase? Most of them coexist in the mixture of polymorphs or stacking faults…

Line 483:

Anna Fontcuberta i Morral's research team advanced in the study of Si(hP4) nanowires synthesized by CVD.

Anna Fontcuberta i Morral's research team keeps ahead in the study of Si(hP4) nanowires synthesized by CVD.

Line 538:

Based on the comprehensive overview of the international research trends…

Based on the international research trends…

Reviewer 2 Report

The manuscript is ready for publication.

Author Response

(The authors gave the same response as above.)
